# Is Input Sparsity Time Possible for Kernel Low-Rank Approximation?

**Cameron Musco**
MIT
cnmusco@mit.edu

**David P. Woodruff**
Carnegie Mellon University
dwoodruf@cs.cmu.edu

## Abstract

Low-rank approximation is a common tool used to accelerate kernel methods: the $n \times n$ kernel matrix $K$ is approximated via a rank-$k$ matrix $\tilde{K}$ which can be stored in much less space and processed more quickly. In this work we study the limits of computationally efficient low-rank kernel approximation. We show that for a broad class of kernels, including the popular Gaussian and polynomial kernels, computing a relative error $k$-rank approximation to $K$ is at least as difficult as multiplying the input data matrix $A \in \mathbb{R}^{n \times d}$ by an arbitrary matrix $C \in \mathbb{R}^{d \times k}$. Barring a breakthrough in fast matrix multiplication, when $k$ is not too large, this requires $\Omega(\mathrm{nnz}(A)k)$ time where $\mathrm{nnz}(A)$ is the number of non-zeros in $A$. This lower bound matches, in many parameter regimes, recent work on subquadratic time algorithms for low-rank approximation of general kernels [MM16, MW17], demonstrating that these algorithms are unlikely to be significantly improved, in particular to $O(\mathrm{nnz}(A))$ input sparsity runtimes. At the same time there is hope: we show for the first time that $O(\mathrm{nnz}(A))$ time approximation is possible for general radial basis function kernels (e.g., the Gaussian kernel) for the closely related problem of low-rank approximation of the kernelized dataset.

## 1 Introduction

The kernel method is a popular technique used to apply linear learning and classification algorithms to datasets with nonlinear structure. Given training input points $a_1, ..., a_n \in \mathbb{R}^d$, the idea is to replace the standard Euclidean dot product $\langle a_i, a_j \rangle = a_i^T a_j$ with the kernel dot product $\psi(a_i, a_j)$, where $\psi : \mathbb{R}^d \times \mathbb{R}^d \to \mathbb{R}^+$ is some positive semidefinite function. Popular kernel functions include e.g., the Gaussian kernel with $\psi(a_i, a_j) = e^{-\|a_i - a_j\|^2 / \sigma}$ for some bandwidth parameter $\sigma$ and the polynomial kernel of degree $q$ with $\psi(a_i, a_j) = (c + a_i^T a_j)^q$ for some parameter $c$.

Throughout this work, we focus on kernels where $\psi(a_i, a_j)$ is a function of the dot products $a_i^T a_i = \|a_i\|^2$, $a_j^T a_j = \|a_j\|^2$, and $a_i^T a_j$. Such functions encompass many kernels used in practice, including the Gaussian kernel, the Laplace kernel, the polynomial kernel, and the Matern kernels.

Letting $\mathcal{F}$ be the reproducing kernel Hilbert space associated with $\psi(\cdot, \cdot)$, we can write $\psi(a_i, a_j) = \langle \phi(a_i), \phi(a_j) \rangle$ where $\phi : \mathbb{R}^d \to \mathcal{F}$ is a typically non-linear *feature map*. We let $\Phi = [\phi(a_1), ..., \phi(a_n)]^T$ denote the kernelized dataset, whose $i^{th}$ row is the kernelized datapoint $\phi(a_i)$.

There is no requirement that $\Phi$ can be efficiently computed or stored – for example, in the case of the Gaussian kernel, $\mathcal{F}$ is an infinite dimensional space. Thus, kernel methods typically work with the kernel matrix $K \in \mathbb{R}^{n \times n}$ with $K_{i,j} = \psi(a_i, a_j)$. We will also sometimes denote $K = \{\psi(a_i, a_j)\}$ to make it clear which kernel function it is generated by. We can equivalently write $K = \Phi \Phi^T$. As long as all operations of an algorithm only access $\Phi$ via the dot products between its rows, they can thus be implemented using just $K$ without explicitly computing the feature map.

Unfortunately computing $K$ is expensive, and a bottleneck for scaling kernel methods to large datasets. For the kernels we consider, where $\psi$ depends on dot products between the input points, we must at least compute the Gram matrix $AA^T$, requiring $\Theta(n^2d)$ time in general. Even if $A$ is sparse, this takes $\Theta(\text{nnz}(A)n)$ time. Storing $K$ then takes $\Theta(n^2)$ space, and processing it for downstream applications like kernel ridge regression and kernel SVM can be even more expensive.

## 1.1 Low-rank kernel approximation

For this reason, a vast body of work studies how to efficiently approximate $K$ via a low-rank surrogate $\tilde{K}$ [SS00, AMS01, WS01, FS02, RR07, ANW14, LSS13, BJ02, DM05, ZTK08, BW09, CKS11, WZ13, GM13]. If $\tilde{K}$ is rank-$k$, it can be stored in factored form in $O(nk)$ space and operated on quickly – e.g., it can be inverted in just $O(nk^2)$ time to solve kernel ridge regression.

One possibility is to set $\tilde{K} = K_k$ where $K_k$ is $K$'s best $k$-rank approximation – the projection onto its top $k$ eigenvectors. $K_k$ minimizes, over all rank-$k$ $\tilde{K}$, the error $\|K - \tilde{K}\|_F$, where $\|M\|_F$ is the Frobenius norm: $(\sum_{i,j} M_{i,j}^2)^{1/2}$. It in fact minimizes error under any unitarily invariant norm, e.g., the popular spectral norm. Unfortunately, $K_k$ is prohibitively expensive to compute, requiring $\Theta(n^3)$ time in practice, or $n^\omega$ in theory using fast matrix multiplication, where $\omega \approx 2.373$ [LG14].

The idea of much prior work on low-rank kernel approximation is to find $\tilde{K}$ which is nearly as good as $K_k$, but can be computed much more quickly. Specifically, it is natural to ask for $\tilde{K}$ fulfilling the following relative error guarantee for some parameter $\epsilon > 0$:

$$\|K - \tilde{K}\|_F \leq (1 + \epsilon)\|K - K_k\|_F. \tag{1}$$

Other goals, such as nearly matching the spectral norm error $\|K - K_k\|$ or approximating $K$ entry-wise have also been considered [RR07, GM13]. Of particular interest to our results is the closely related goal of outputting an orthonormal basis $Z \in \mathbb{R}^{n \times k}$ satisfying for any $\Phi$ with $\Phi\Phi^T = K$:

$$\|\Phi - ZZ^T\Phi\|_F \leq (1 + \epsilon)\|\Phi - \Phi_k\|_F. \tag{2}$$

(2) can be viewed as a Kernel PCA guarantee – its asks us to find a low-rank subspace $Z$ such that the projection of our kernelized dataset $\Phi$ onto $Z$ nearly optimally approximates this dataset. Given $Z$, we can approximate $K$ using $\tilde{K} = ZZ^T\Phi\Phi^TZZ^T = ZZ^TKZZ^T$. Alternatively, letting $P$ be the projection onto the row span of $ZZ^T\Phi$, we can write $\tilde{K} = \Phi P\Phi^T$, which can be computed efficiently, for example, when $P$ is a projection onto a subset of the kernelized datapoints [MM16].

## 1.2 Fast algorithms for relative-error kernel approximation

Until recently, all algorithms achieving the guarantees of (1) and (2) were at least as expensive as computing the full matrix $K$, which was needed to compute the low-rank approximation [GM13].

However, recent work has shown that this is not required. Avron, Nguyen, and Woodruff [ANW14] demonstrate that for the polynomial kernel, $Z$ satisfying (2) can be computed in $O(\text{nnz}(A)q) + n\,\text{poly}(3^qk/\epsilon)$ time for a polynomial kernel with degree $q$.

Musco and Musco [MM16] give a fast algorithm for *any kernel*, using recursive Nyström sampling, which computes $\tilde{K}$ (in factored form) satisfying $\|K - \tilde{K}\| \leq \lambda$, for input parameter $\lambda$. With the proper setting of $\lambda$, it can output $Z$ satisfying (2) (see Section C.3 of [MM16]). Computing $Z$ requires evaluating $\tilde{O}(k/\epsilon)$ columns of the kernel matrix along with $\tilde{O}(n(k/\epsilon)^{\omega-1})$ additional time for other computations. Assuming the kernel is a function of the dot products between the input points, the kernel evaluations require $\tilde{O}(\text{nnz}(A)k/\epsilon)$ time. The results of [MM16] can also be used to compute $\tilde{K}$ satisfying (1) with $\epsilon = \sqrt{n}$ in $\tilde{O}(\text{nnz}(A)k + nk^{\omega-1})$ time (see Appendix A of [MW17]).

Woodruff and Musco [MW17] show that for any kernel, and for any $\epsilon > 0$, it is possible to achieve (1) in $\tilde{O}(\text{nnz}(A)k/\epsilon) + n\,\text{poly}(k/\epsilon)$ time plus the time needed to compute an $\tilde{O}(\sqrt{nk}/\epsilon^2) \times \tilde{O}(\sqrt{nk}/\epsilon)$ submatrix of $K$. If $A$ has uniform row sparsity – i.e., $\text{nnz}(a_i) \leq c\,\text{nnz}(A)/n$ for some constant $c$ and all $i$, this step can be done in $\tilde{O}(\text{nnz}(A)k/\epsilon^{2.5})$ time. Alternatively, if $d \leq (\sqrt{nk}/\epsilon^2)^\alpha$ for $\alpha < .314$ this can be done in $\tilde{O}(nk/\epsilon^4) = \tilde{O}(\text{nnz}(A)k/\epsilon^4)$ time using fast rectangular matrix multiplication [LG12, GU17] (assuming that there are no all zero data points so $n \leq \text{nnz}(A)$.)

## 1.3 Our results

The algorithms of [MM16, MW17] make significant progress in efficiently solving (1) and (2) for general kernel matrices. They demonstrate that, surprisingly, a relative-error low-rank approximation can be computed significantly faster than the time required to write down all of $K$.

A natural question is if these results can be improved. Even ignoring $\epsilon$ dependencies and typically lower order terms, both algorithms use $\Omega(\mathrm{nnz}(A)k)$ time. One might hope to improve this to input sparsity, or near input sparsity time, $\tilde{O}(\mathrm{nnz}(A))$, which is known for computing a low-rank approximation of $A$ itself [CW13]. The work of Avron et al. affirms that this is possible for the kernel PCA guarantee of (2) for degree-$q$ polynomial kernels, for constant $q$. Can this result be extended to other popular kernels, or even more general classes?

### 1.3.1 Lower bounds

We show that achieving the guarantee of (1) significantly more efficiently than the work of [MM16, MW17] is likely very difficult. Specifically, we prove that for a wide class of kernels, the kernel low-rank approximation problem is as hard as multiplying the input $A \in \mathbb{R}^{n \times d}$ by an arbitrary $C \in \mathbb{R}^{d \times k}$. We have the following result for some common kernels to which our techniques apply:

**Theorem 1** (Hardness for low-rank kernel approximation)**.** *Consider any polynomial kernel $\psi(m_i, m_j) = (c + m_i^T m_j)^q$, Gaussian kernel $\psi(m_i, m_j) = e^{-\|m_i - m_j\|^2/\sigma}$, or the linear kernel $\psi(m_i, m_j) = m_i^T m_j$. Assume there is an algorithm which given $M \in \mathbb{R}^{n \times d}$ with associated kernel matrix $K = \{\psi(m_i, m_j)\}$, returns $N \in \mathbb{R}^{n \times k}$ in $o(\mathrm{nnz}(M)k)$ time satisfying:*

$$\|K - NN^T\|_F^2 \le \Delta \|K - K_k\|_F^2$$

*for some approximation factor $\Delta$. Then there is an $o(\mathrm{nnz}(A)k) + O(nk^2)$ time algorithm for multiplying arbitrary integer matrices $A \in \mathbb{R}^{n \times d}$, $C \in \mathbb{R}^{d \times k}$.*

The above applies for *any approximation factor* $\Delta$. While we work in the real RAM model, ignoring bit complexity, as long as $\Delta = \mathrm{poly}(n)$ and $A, C$ have polynomially bounded entries, our reduction from multiplication to low-rank approximation is achieved using matrices that can be represented with just $O(\log(n + d))$ bits per entry.

Theorem 1 shows that the runtime of $\tilde{O}(\mathrm{nnz}(A)k + nk^{\omega-1})$ for $\Delta = \sqrt{n}$ achieved by [MM16] for general kernels cannot be significantly improved without advancing the state-of-the-art in matrix multiplication. Currently no general algorithm is known for multiplying integer $A \in \mathbb{R}^{n \times d}$, $C \in \mathbb{R}^{d \times k}$ in $o(\mathrm{nnz}(A)k)$ time, except when $k \ge n^\alpha$ for $\alpha < .314$ and $A$ is dense. In this case, $AC$ can be computed in $O(nd)$ time using fast rectangular matrix multiplication [LG12, GU17].

As discussed, when $A$ has uniform row sparsity or when $d \le (\sqrt{nk}/\epsilon^2)^\alpha$, the runtime of [MW17] for $\Delta = (1 + \epsilon)$, ignoring $\epsilon$ dependencies and typically lower order terms, is $\tilde{O}(\mathrm{nnz}(A)k)$, which is also nearly tight.

In recent work, Backurs et al. [BIS17] give lower bounds for a number of kernel learning problems, including kernel PCA for the Gaussian kernel. However, their strong bound, of $\Omega(n^2)$ time, requires very small error $\Delta = \exp(-\omega(\log^2 n)$, whereas ours applies for any relative error $\Delta$.

### 1.3.2 Improved algorithm for radial basis function kernels

In contrast to the above negative result, we demonstrate that achieving the alternative Kernel PCA guarantee of (2) is possible in input sparsity time for any shift and rotationally invariant kernel – e.g., any radial basis function kernel where $\psi(x_i, x_j) = f(\|x_i - x_j\|)$. This result significantly extends the progress of Avron et al. [ANW14] on the polynomial kernel.

Our algorithm is based off a fast implementation of the random Fourier features method [RR07], which uses the fact that that the Fourier transform of any shift invariant kernel is a probability distribution after appropriate scaling (a consequence of Bochner's theorem). Sampling frequencies from this distribution gives an approximation to $\psi(\cdot, \cdot)$ and consequentially the matrix $K$.

We employ a new analysis of this method [AKM+17], which shows that sampling $\tilde{O}\left(\frac{n}{\epsilon^2\lambda}\right)$ random Fourier features suffices to give $\tilde{K} = \tilde{\Phi}\tilde{\Phi}^T$ satisfying the spectral approximation guarantee:

$$(1-\epsilon)(\tilde{K}+\lambda I) \preceq K + \lambda I \preceq (1+\epsilon)(\tilde{K}+\lambda I).$$

If we set $\lambda \leq \sigma_{k+1}(K)/k$, we can show that $\tilde{\Phi}$ also gives a *projection-cost preserving sketch* [CEM+15] for the kernelized dataset $\Phi$. This ensures that any $Z$ satisfying $\|\tilde{\Phi} - ZZ^T\tilde{\Phi}\|_F^2 \leq (1+\epsilon)\|\tilde{\Phi} - \tilde{\Phi}_k\|_F^2$ also satisfies $\|\Phi - ZZ^T\Phi\|_F^2 \leq (1+O(\epsilon))\|\Phi - \Phi_k\|_F^2$ and thus achieves (2).

Our algorithm samples $s = \tilde{O}\left(\frac{n}{\epsilon^2\lambda}\right) = \tilde{O}\left(\frac{nk}{\epsilon^2\sigma_{k+1}(K)}\right)$ random Fourier features, which naively requires $O(\mathrm{nnz}(A)s)$ time. We show that this can be accelerated to $O(\mathrm{nnz}(A)) + \mathrm{poly}(n, s)$ time, using a recent result of Kapralov et al. on fast multiplication by random Gaussian matrices [KPW16]. Our technique is analogous to the 'Fastfood' approach to accelerating random Fourier features using fast Hadamard transforms [LSS13]. However, our runtime scales with $\mathrm{nnz}(A)$, which can be significantly smaller than the $\tilde{O}(nd)$ runtime given by Fastfood when $A$ is sparse. Our main algorithmic result is:

**Theorem 2** (Input sparsity time kernel PCA). *There is an algorithm that given $A \in \mathbb{R}^{n \times d}$ along with shift and rotation-invariant kernel function $\psi : \mathbb{R}^d \times \mathbb{R}^d \to \mathbb{R}^+$ with $\psi(x,x) = 1$, outputs, with probability $99/100$, $Z \in \mathbb{R}^{n \times k}$ satisfying:*

$$\|\Phi - ZZ^T\Phi\|_F^2 \leq (1+\epsilon)\|\Phi - \Phi_k\|_F^2$$

*for any $\Phi$ with $\Phi\Phi^T = K = \{\psi(a_i, a_j)\}$ and any $\epsilon > 0$. Letting $\sigma_{k+1}$ denote the $(k+1)^{th}$ largest eigenvalue of $K$ and $\omega < 2.373$ be the exponent of fast matrix multiplication, the algorithm runs in $O(\mathrm{nnz}(A)) + \tilde{O}\left(n^{\omega+1.5} \cdot \left(\frac{k}{\sigma_{k+1}\epsilon^2}\right)^{\omega-1.5}\right)$ time.*

We note that the runtime of our algorithm is $O(\mathrm{nnz}(A))$ whenever $n$, $k$, $1/\sigma_{k+1}$, and $1/\epsilon$ are not too large. Due to the relatively poor dependence on $n$, the algorithm is relevant for very high dimensional datasets with $d \gg n$. Such datasets are found often, e.g., in genetics applications [HDC+01, JDMP11]. While we have dependence on $1/\sigma_{k+1}$, in the natural setting, we only compute a low-rank approximation up to an error threshold, ignoring very small eigenvalues of $K$, and so $\sigma_{k+1}$ will not be too small. We do note that if we apply Theorem 2 to the low-rank approximation instances given by our lower bound construction, $\sigma_{k+1}$ can be very small, $\leq 1/\mathrm{poly}(n,d)$ for matrices with $\mathrm{poly}(n)$ bounded entries. Thus, removing this dependence is an important open question in understanding the complexity of low-rank kernel approximation.

We leave open the possibility of improving our algorithm, achieving $O(\mathrm{nnz}(A)) + n \cdot \mathrm{poly}(k, \epsilon)$ runtime, which would match the state-of-the-art for low-rank approximation of non-kernelized matrices [CW13]. Alternatively, it is possible that a lower bound can be shown, proving the that high $n$ dependence, or the $1/\sigma_{k+1}$ term are required even for the Kernel PCA guarantee of (2).

## 2 Lower bounds

Our lower bound proof argues that for a broad class of kernels, given input $M$, a low-rank approximation of the associated kernel matrix $K$ achieving (1) can be used to obtain a close approximation to the Gram matrix $MM^T$. We write $\psi(m_i^T m_j)$ as a function of $m_i^T m_j$ (or $\|m_i - m_j\|^2$ for distance kernels) and expand this function as a power series. We show that the if input points are appropriately rescaled, the contribution of degree-1 term $m_i^T m_j$ dominates, and hence our kernel matrix approximates $MM^T$, up to some easy to compute low-rank components.

We then show that such an approximation can be used to give a fast algorithm for multiplying any two integer matrices $A \in \mathbb{R}^{n \times d}$ and $C \in \mathbb{R}^{d \times k}$. The key idea is to set $M = [A, wC]$ where $w$ is a large weight. We then have:

$$MM^T = \begin{bmatrix} AA^T & wAC \\ wC^T A^T & w^2 C^T C \end{bmatrix}.$$

Since w is very large, the $AA^T$ block is relatively very small, and so $MM^T$ is nearly rank-$2k$ – it has a 'heavy' strip of elements in its last $k$ rows and columns. Thus, computing a relative-error rank-$2k$ approximation to $MM^T$ recovers all entries except those in the $AA^T$ block very accurately, and importantly, recovers the $wAC$ block and so the product $AC$.

## 2.1 Lower bound for low-rank approximation of $MM^T$.

We first illustrate our lower bound technique by showing hardness of direct approximation of $MM^T$.

**Theorem 3** (Hardness of low-rank approximation for $MM^T$). *Assume there is an algorithm $\mathcal{A}$ which given any $M \in \mathbb{R}^{n \times d}$ returns $N \in \mathbb{R}^{n \times k}$ such that $\|MM^T - NN^T\|_F^2 \leq \Delta_1\|MM^T - (MM^T)_k\|_F^2$ in $T(M,k)$ time for some approximation factor $\Delta_1$.*

*For any $A \in \mathbb{R}^{n \times d}$ and $C \in \mathbb{R}^{d \times k}$ each with integer entries in $[-\Delta_2, \Delta_2]$, let $B = [A^T, wC]^T$ where $w = 3\sqrt{\Delta_1}\Delta_2^2 nd$. It is possible to compute the product $AC$ in time $T(B, 2k) + O(nk^{\omega-1})$.*

*Proof.* We can write the $(n+k) \times (n+k)$ matrix $BB^T$ as:

$$BB^T = [A^T, wC]^T[A, wC] = \begin{bmatrix} AA^T & wAC \\ wC^TA^T & w^2C^TC \end{bmatrix}.$$

Let $Q \in \mathbb{R}^{n \times 2k}$ be an orthogonal span for the columns of the $n \times 2k$ matrix:

$$\begin{bmatrix} 0 & wAC \\ V & w^2C^TC \end{bmatrix}$$

where $V \in \mathbb{R}^{k \times k}$ spans the columns of $wC^TA^T \in \mathbb{R}^{k \times n}$. The projection $QQ^TBB^T$ gives the best Frobenius norm approximation to $BB^T$ in the span of $Q$. We can see that:

$$\|BB^T - (BB^T)_{2k}\|_F^2 \leq \|BB^T - QQ^TBB^T\|_F^2 \leq \left\| \begin{bmatrix} AA^T & 0 \\ 0 & 0 \end{bmatrix} \right\|_F^2 \leq \Delta_2^4 n^2 d^2 \qquad (3)$$

since each entry of $A$ is bounded in magnitude by $\Delta_2$ and so each entry of $AA^T$ is bounded by $d\Delta_2^2$.

Let $N$ be the matrix returned by running $\mathcal{A}$ on $B$ with rank $2k$. In order to achieve the approximation bound of $\|BB^T - NN^T\|_F^2 \leq \Delta_1\|BB^T - (BB^T)_{2k}\|_F^2$ we must have, for all $i, j$:

$$(BB^T - NN^T)_{i,j}^2 \leq \|BB^T - NN^T\|_F^2 \leq \Delta_1\Delta_2^4 n^2 d^2$$

where the last inequality is from (3). This gives $|BB^T - NN^T|_{i,j} \leq \sqrt{\Delta_1}\Delta_2^2 nd$. Since $A$ and $C$ have integer entries, each entry in the submatrix $wAC$ of $BB^T$ is an integer multiple of $w = 3\sqrt{\Delta_1}\Delta_2^2 nd$. Since $(NN^T)_{i,j}$ approximates this entry to error $\sqrt{\Delta_1}\Delta_2^2 nd$, by simply rounding $(NN^T)_{i,j}$ to the nearest multiple of $w$, we obtain the entry exactly. Thus, given $N$, we can exactly recover $AC$ in $O(nk^{\omega-1})$ time by computing the $n \times k$ submatrix corresponding to $AC$ in $BB^T$. $\square$

Theorem 3 gives our main bound Theorem 1 for the case of the linear kernel $\psi(m_i, m_j) = m_i^T m_j$.

*Proof of Theorem 1 – Linear Kernel.* We apply Theorem 3 after noting that for $B = [A^T, wC]^T$, $\text{nnz}(B) \leq \text{nnz}(A) + nk$ and so $T(B, 2k) = o(\text{nnz}(A)k) + O(nk^2)$. $\square$

We show in Appendix A that there is an algorithm which nearly matches the lower bound of Theorem 1 for any $\Delta = (1 + \epsilon)$ for any $\epsilon > 0$. Further, in Appendix B we show that even just outputting an orthogonal matrix $Z \in \mathbb{R}^{n \times k}$ such that $\tilde{K} = ZZ^TMM^T$ is a relative-error low-rank approximation of $MM^T$, but not computing a factorization of $\tilde{K}$ itself, is enough to give fast multiplication of integer matrices $A$ and $C$.

## 2.2 Lower bound for dot product kernels

We now extend Theorem 3 to general dot product kernels – where $\psi(a_i, a_j) = f(a_i^T a_j)$ for some function $f$. This includes, for example, the polynomial kernel.

**Theorem 4** (Hardness of low-rank approximation for dot product kernels). *Consider any kernel $\psi : \mathbb{R}^d \times \mathbb{R}^d \to \mathbb{R}^+$ with $\psi(a_i, a_j) = f(a_i^T a_j)$ for some function $f$ which can be expanded as $f(x) = \sum_{q=0}^{\infty} c_q x^q$ with $c_1 \neq 0$ and $|c_q/c_1| \leq G^{q-1}$ and for all $q \geq 2$ and some $G \geq 1$.*

*Assume there is an algorithm $\mathcal{A}$ which given $M \in \mathbb{R}^{n \times d}$ with kernel matrix $K = \{\psi(m_i, m_j)\}$, returns $N \in \mathbb{R}^{n \times k}$ satisfying $\|K - NN^T\|_F^2 \leq \Delta_1\|K - K_k\|$ in $T(M,k)$ time.*

*For any $A \in \mathbb{R}^{n \times d}$, $C \in \mathbb{R}^{d \times k}$ with integer entries in $[-\Delta_2, \Delta_2]$, let $B = [w_1 A^T, w_2 C]^T$ with $w_1 = \frac{w_2}{12\sqrt{\Delta_1}\Delta_2^2 nd}$, $w_2 = \frac{1}{4\sqrt{Gd}\Delta_2}$. Then it is possible to compute $AC$ in time $T(B, 2k+1) + O(nk^{\omega-1})$.*

*Proof.* Using our decomposition of $\psi(\cdot, \cdot)$, we can write the kernel matrix for $B$ and $\psi$ as:

$$K = c_0 \begin{bmatrix} 1 & 1 \\ 1 & 1 \end{bmatrix} + c_1 \begin{bmatrix} w_1^2 AA^T & w_1 w_2 AC \\ w_1 w_2 C^T A^T & w_2^2 C^T C \end{bmatrix} + c_2 K^{(2)} + c_3 K^{(3)} + ... \tag{4}$$

where $K_{i,j}^{(q)} = (b_i^T b_j)^q$ and $1$ denotes the all ones matrix of appropriate size. The key idea is to show that the contribution of the $K^{(q)}$ terms is small, and so any relative-error rank-$(2k+1)$ approximation to $K$ must recover an approximation to $BB^T$, and thus the product $AC$ as in Theorem 3.

By our setting of $w_2 = \frac{1}{4\sqrt{G}d\Delta_2}$, the fact that $w_1 < w_2$, and our bound on the entries of $A$ and $C$, we have for all $i, j$, $|b_i^T b_j| \leq w_2^2 d\Delta_2^2 < \frac{1}{16G}$. Thus, for any $i, j$, using that $|c_q/c_1| \leq G^{q-1}$:

$$\left| \sum_{q=2}^{\infty} c_q K_{i,j}^{(q)} \right| \leq c_1 |b_i^T b_j| \cdot \left| \sum_{q=2}^{\infty} G^{q-1} |b_i^T b_j|^{q-1} \right| \leq c_1 |b_i^T b_j| \sum_{q=2}^{\infty} \frac{G^{q-1}}{(16G)^{q-1}} \leq \frac{1}{12} c_1 |b_i^T b_j|. \tag{5}$$

Let $\bar{K}$ be the matrix $\left( K - c_0 \begin{bmatrix} 1 & 1 \\ 1 & 1 \end{bmatrix} \right)$, with its top right $n \times n$ block set to 0. $\bar{K}$ just has its last $k$ columns and rows non-zero, so has rank $\leq 2k$. Let $Q \in \mathbb{R}^{n \times 2k+1}$ be an orthogonal span for the columns $\bar{K}$ along with the all ones vector of length $n$. Let $N$ be the result of running $\mathcal{A}$ on $B$ with rank $2k+1$. Then we have:

$$\|K - NN^T\|_F^2 \leq \Delta_1 \|K - K_{2k+1}\|_F^2 \leq \Delta_1 \|K - QQ^T K\|_F^2$$
$$\leq \Delta_1 \left\| \begin{bmatrix} (c_1 w_1^2 AA^T + c_2 \hat{K}^{(2)} + ...) & 0 \\ 0 & 0 \end{bmatrix} \right\|_F^2 \tag{6}$$

where $\hat{K}^{(q)}$ denotes the top left $n \times n$ submatrix of $K^{(q)}$. By our bound on the entries of $A$ and (5):

$$\left| \left( c_1 w_1^2 AA^T + c_2 \hat{K}^{(2)} + c_3 \hat{K}^{(3)} + ... \right)_{i,j} \right| \leq \frac{13}{12} \left| \left( c_1 w_1^2 AA^T \right)_{i,j} \right| \leq 2 c_1 w_1^2 d \Delta_2^2.$$

Plugging back into (6) and using $w_1 = \frac{w_2}{12\sqrt{\Delta_1 \Delta_2^2 nd}}$, this gives for any $i, j$:

$$(K - NN^T)_{i,j} \leq \|K - NN^T\|_F \leq \sqrt{\Delta_1 n^2} \cdot 2 c_1 w_1^2 d \Delta_2^2$$
$$\leq \frac{\sqrt{\Delta_1} n \cdot 2 c_1 d \Delta_2^2}{12\sqrt{\Delta_1} \Delta_2^2 nd} \cdot w_1 w_2$$
$$\leq \frac{w_1 w_2 c_1}{6}. \tag{7}$$

Since $A$ and $C$ have integer entries, each entry of $c_1 w_1 w_2 AC$ is an integer multiple of $c_1 w_1 w_2$. By the decomposition of (4) and the bound of (5), if we subtract $c_0$ from the corresponding entry of $K$ and round it to the nearest multiple of $c_1 w_1 w_2$, we will recover the entry of $AC$. By the bound of (7), we can likewise round the corresponding entry of $NN^T$. Computing all $nk$ of these entries given $N$ takes time $O(nk^{\omega-1})$, giving the theorem. $\square$

Theorem 4 lets us lower bound the time to compute a low-rank kernel approximation for any kernel function expressible as a reasonable power expansion of $a_i^T a_j$. As a straightforward example, it gives the lower bound for the polynomial kernel of any degree stated in Theorem 1.

*Proof of Theorem 1 – Polynomial Kernel.* We apply Theorem 4, noting that $\psi(m_i, m_j) = (c + m_i^T m_j)^q$ can be written as $f(m_i^T m_j)$ where $f(x) = \sum_{j=0}^{q} c_j x^j$ with $c_j = c^{q-j} \binom{q}{j}$. Thus $c_1 \neq 0$ and $|c_j/c_1| \leq G^{j-1}$ for $G = (q/c)$. Finally note that $\text{nnz}(B) \leq \text{nnz}(A) + nk$ giving the result. $\square$

## 2.3 Lower bound for distance kernels

We finally extend Theorem 4 to handle kernels like the Gaussian kernel whose value depends on the squared distance $\|a_i - a_j\|^2$ rather than just the dot product $a_i^T a_j$. We prove:

**Theorem 5** (Hardness of low-rank approximation for distance kernels). *Consider any kernel function $\psi : \mathbb{R}^d \times \mathbb{R}^d \to \mathbb{R}^+$ with $\psi(a_i, a_j) = f(\|a_i - a_j\|^2)$ for some function $f$ which can be expanded as $f(x) = \sum_{q=0}^{\infty} c_q x^q$ with $c_1 \neq 0$ and $|c_q/c_1| \leq G^{q-1}$ and for all $q \geq 2$ and some $G \geq 1$.*

*Assume there is an algorithm $\mathcal{A}$ which given input $M \in \mathbb{R}^{n \times d}$ with kernel matrix $K = \{\psi(m_i, m_j)\}$, returns $N \in \mathbb{R}^{n \times k}$ satisfying $\|K - NN^T\|_F^2 \leq \Delta_1 \|K - K_k\|$ in $T(M, k)$ time.*

*For any $A \in \mathbb{R}^{n \times d}$, $C \in \mathbb{R}^{d \times k}$ with integer entries in $[-\Delta_2, \Delta_2]$, let $B = [w_1 A^T, w_2 C]^T$ with $w_1 = \frac{w_2}{36\sqrt{\Delta_1}\Delta_2^2 nd}$, $w_2 = \frac{1}{(16Gd^2\Delta_2^4)(36\sqrt{\Delta_1}\Delta_2^2 nd)}$. It is possible to compute $AC$ in $T(B, 2k+3) + O(nk^{\omega-1})$ time.*

The proof of Theorem 5 is similar to that of Theorem 4, and relegated to Appendix C. The key idea is to write $K$ as a polynomial in the *distance matrix* $D$ with $D_{i,j} = \|b_i - b_j\|_2^2$. Since $\|b_i - b_j\|_2^2 = \|b_i\|_2^2 + \|b_j\|_2^2 - 2b_i^T b_j$, $D$ can be written as $-2BB^T$ plus a rank-2 component. By setting $w_1, w_2$ sufficiently small, as in the proof of Theorem 4, we ensure that the higher powers of $D$ are negligible, and thus that our low-rank approximation must accurately recover the submatrix of $BB^T$ corresponding to $AC$. Theorem 5 gives Theorem 1 for the popular Gaussian kernel:

*Proof of Theorem 1 – Gaussian Kernel.* $\psi(m_i, m_j)$ can be written as $f(\|m_i - m_j\|^2)$ where $f(x) = e^{-x/\sigma} = \sum_{q=0}^{\infty} \frac{(-1/\sigma)^q}{q!} x^q$. Thus $c_1 \neq 0$ and $|c_q/c_1| \leq G^{q-1}$ for $G = 1/\sigma$. Applying Theorem 5 and bounding $\text{nnz}(B) \leq \text{nnz}(A) + nk$, gives the result. □

# 3  Input sparsity time kernel PCA for radial basis kernels

Theorem 1 gives little hope for achieving $o(\text{nnz}(A)k)$ time for low-rank kernel approximation. However, the guarantee of (1) is not the only way of measuring the quality of $\tilde{K}$. Here we show that for shift/rotationally invariant kernels, including e.g., radial basis kernels, input sparsity time can be achieved for the kernel PCA goal of (2).

## 3.1  Basic algorithm

Our technique is based on the random Fourier features technique [RR07]. Given any shift-invariant kernel, $\psi(x, y) = \psi(x - y)$ with $\psi(0) = 1$ (we will assume this w.l.o.g. as the function can always be scaled), there is a probability density function $p(\eta)$ over vectors in $\mathbb{R}^d$ such that:

$$\psi(x - y) = \int_{\mathbb{R}^d} e^{-2\pi i \eta^T (x-y)} p(\eta) d\eta. \tag{8}$$

$p(\eta)$ is just the (inverse) Fourier transform of $\psi(\cdot)$, and is a density function by Bochner's theorem. Informally, given $A \in \mathbb{R}^{n \times d}$ if we let $Z$ denote the matrix with columns $z(\eta)$ indexed by $\eta \in \mathbb{R}^d$. $z(\eta)_j = e^{-2\pi i \eta^T a_j}$. Then (8) gives $ZPZ^* = K$ where $P$ is diagonal with $P_{\eta,\eta} = p(\eta)$, and $Z^*$ denotes the Hermitian transpose.

The idea of random Fourier features is to select $s$ frequencies $\eta_1, ..., \eta_s$ according to the density $p(\eta)$ and set $\tilde{Z} = \frac{1}{\sqrt{s}}[z(\eta_1), ...z(\eta_s)]$. $\tilde{K} = \tilde{Z}\tilde{Z}^T$ is then used to approximate $K$.

In recent work, Avron et al. [AKM$^+$17] give a new analysis of random Fourier features. Extending prior work on ridge leverage scores in the discrete setting [AM15, CMM17], they define the *ridge leverage function* for parameter $\lambda > 0$:

$$\tau_\lambda(\eta) = p(\eta)z(\eta)^*(K + \lambda I)^{-1} z(\eta) \tag{9}$$

As part of their results, which seek $\tilde{K}$ that spectrally approximates $K$, they prove the following:

**Lemma 6.** *For all $\eta$, $\tau_\lambda(\eta) \leq n/\lambda$.*

While simple, this bound is key to our algorithm. It was shown in [CMM17] that if the columns of a matrix are sampled by over-approximations to their ridge leverage scores (with appropriately set $\lambda$), the sample is a projection-cost preserving sketch for the original matrix. That is, it can be used as a surrogate in computing a low-rank approximation. The results of [CMM17] carry over to the continuous setting giving, in conjunction with Lemma 6:

**Lemma 7** (Projection-cost preserving sketch via random Fourier features). *Consider any $A \in \mathbb{R}^{n \times d}$ and shift-invariant kernel $\psi(\cdot)$ with $\psi(0) = 1$, with associated kernel matrix $K = \{\psi(a_i - a_j)\}$ and kernel Fourier transform $p(\eta)$. For any $0 < \lambda \leq \frac{1}{k} \sum_{i=k+1}^{n} \sigma_i(K)$, let $s = \frac{cn \log(n/\delta\lambda)}{\epsilon^2 \lambda}$ for sufficiently large $c$ and let $\tilde{Z} = \frac{1}{\sqrt{s}}[z(\eta_1), ..., z(\eta_s)]$ where $\eta_1, ..., \eta_s$ are sampled independently according to $p(\eta)$. Then with probability $\geq 1 - \delta$, for any orthonormal $Q \in \mathbb{R}^{n \times k}$ and any $\Phi$ with $\Phi\Phi^T = K$:*

$$(1 - \epsilon)\|QQ^T\tilde{Z} - \tilde{Z}\|_F^2 \leq \|QQ^T\Phi - \Phi\|_F^2 \leq (1 + \epsilon)\|QQ^T\tilde{Z} - \tilde{Z}\|_F^2. \tag{10}$$

By (10) if we compute $Q$ satisfying $\|QQ^T\tilde{Z} - \tilde{Z}\|_F^2 \leq (1 + \epsilon)\|\tilde{Z} - \tilde{Z}_k\|_F^2$ then we have:

$$\|QQ^T\Phi - \Phi\|_F^2 \leq (1 + \epsilon)^2\|\tilde{Z} - \tilde{Z}_k\|_F^2 \leq \frac{(1 + \epsilon)^2}{1 - \epsilon}\|U_k U_k^T\Phi - \Phi\|_F^2$$
$$= (1 + O(\epsilon))\|\Phi - \Phi_k\|_F^2$$

where $U_k \in \mathbb{R}^{n \times k}$ contains the top $k$ column singular vectors of $\Phi$. By adjusting constants on $\epsilon$ by making $c$ large enough, we thus have the relative error low-rank approximation guarantee of (2). It remains to show that this approach can be implemented efficiently.

## 3.2 Input sparsity time implementation

Given $\tilde{Z}$ sampled as in Lemma 7, we can find a near optimal subspace $Q$ using any input sparsity time low-rank approximation algorithm (e.g., [CW13, NN13]). We have the following Corollary:

**Corollary 8.** *Given $\tilde{Z}$ sampled as in Lemma 7 with $s = \tilde{\Theta}(\frac{nk}{\epsilon^2 \sigma_{k+1}(K)})$, there is an algorithm running in time $\tilde{O}(\frac{n^2 k}{\epsilon^2 \sigma_{k+1}(K)})$ that computes $Q$ satisfying with high probability, for any $\Phi$ with $\Phi\Phi^T = K$:*

$$\|QQ^T\Phi - \Phi\|_F^2 \leq (1 + \epsilon)\|\Phi - \Phi_k\|_F^2.$$

With Corollary 8 in place the main bottleneck to our approach becomes computing $\tilde{Z}$.

### 3.2.1 Sampling Frequencies

To compute $\tilde{Z}$, we first sample $\eta_1, ..., \eta_s$ according to $p(\eta)$. Here we use the rotational invariance of $\psi(\cdot)$. In this case, $p(\eta)$ is also rotationally invariant [LSS13] and so, letting $\hat{p}(\cdot)$ be the distribution over norms of vectors sampled from $p(\eta)$ we can sample $\eta_1, ..., \eta_n$ by first selecting $s$ random Gaussian vectors and then rescaling them to have norms distributed according to $\hat{p}(\cdot)$. That is, we can write $[\eta_1, ..., \eta_n] = GD$ where $G \in \mathbb{R}^{d \times s}$ is a random Gaussian matrix and $D$ is a diagonal rescaling matrix with $D_{ii} = \frac{m}{\|G_i\|}$ with $m \sim \hat{p}$. We will assume that $\hat{p}$ can be sampled from in $O(1)$ time. This is true for many natural kernels – e.g., for the Gaussian kernel, $\hat{p}$ is just a Gaussian density.

### 3.2.2 Computing $\tilde{Z}$

Due to our large sample size, $s > n$, even writing down $G$ above requires $\Omega(nd)$ time. However, to form $\tilde{Z}$ we do not need $G$ itself: it suffices to compute for $m = 1, ..., s$ the column $z(\eta_m)$ with $z(\eta_m)_j = e^{-2\pi i \eta_m^T a_j}$. This requires computing $AGD$, which contains the appropriate dot products $a_j^T \eta_m$ for all $m, j$. We use a recent result [KPW16] which shows that this can be performed approximately in input sparsity time:

**Lemma 9** (From Theorem 1 of [KPW16]). *There is an algorithm running in $O(\text{nnz}(A) + \frac{\log^4 dn^3 s^{\omega-1.5}}{\delta})$ time which outputs random $B$ whose distribution has total variation distance at most $\delta$ from the distribution of $AG$ where $G \in \mathbb{R}^{d \times s}$ is a random Gaussian matrix. Here, $\omega < 2.373$ is the exponent of fast matrix multiplication.*

*Proof.* Theorem 1 of [KPW16] shows that for $B$ to have total variation distance $\delta$ from the distribution of $AG$ it suffices to set $B = ACG'$ where $C$ is a $d \times O(\log^4 dn^2 s^{1/2}/\delta)$ CountSketch matrix

and $G'$ is an $O(\log^4 dn^2 s^{1/2}/\delta) \times s$ random Gaussian matrix. Computing $AC$ requires $O(\mathrm{nnz}(A))$ time. Multiplying the result by $G'$ then requires $O(\frac{\log^4 dn^3 s^{1.5}}{\delta})$ time if fast matrix multiplication is not employed. Using fast matrix multiplication, this can be improved to $O(\frac{\log^4 dn^3 s^{\omega-1.5}}{\delta})$. $\qquad\square$

Applying Lemma 9 with $\delta = 1/200$ lets us compute random $BD$ with total variation distance $1/200$ from $AGD$. Thus, the distribution of $\tilde{Z}$ generated from this matrix has total variation distance $\leq 1/200$ from the $\tilde{Z}$ generated from the true random Fourier features distribution. So, by Corollary 8, we can use $\tilde{Z}$ to compute $Q$ satisfying $\|QQ^T\Phi - \Phi\|_F^2 \leq (1+\epsilon)\|\Phi - \Phi_k\|_F^2$ with probability $1/100$ accounting for the the total variation difference and the failure probability of Corollary 8. This yields our main algorithmic result, Theorem 2.

### 3.3 An alternative approach

We conclude by noting that near input sparsity time Kernel PCA can also be achieved for a broad class of kernels using a very different approach. We can approximate $\psi(\cdot,\cdot)$ via an expansion into polynomial kernel matrices as is done in [CKS11] and then apply the sketching algorithms for the polynomial kernel developed in [ANW14]. As long as the expansion achieves high accuracy with low degree, and as long as $1/\sigma_{k+1}$ is not too small – since this will control the necessary approximation factor, this technique can yield runtimes of the form $\tilde{O}(\mathrm{nnz}(A)) + \mathrm{poly}(n, k, 1/\sigma_{k+1}, 1/\epsilon)$, giving improved dependence on $n$ for some kernels over our random Fourier features method. Improving the $\mathrm{poly}(n, k, 1/\sigma_{k+1}, 1/\epsilon)$ term in both these methods, and especially removing the $1/\sigma_{k+1}$ dependence and achieving linear dependence on $n$ is an interesting open question for future work.

## 4 Conclusion

In this work we have shown that for a broad class of kernels, including the Gaussian, polynomial, and linear kernels, given data matrix $A$, computing a relative error low-rank approximation to $A$'s kernel matrix $K$ (i.e., satisfying (1)) requires at least $\Omega(\mathrm{nnz}(A)k)$ time, barring a major breakthrough in the runtime of matrix multiplication. In the constant error regime, this lower bound essentially matches the runtimes given by recent work on subquadratic time kernel and PSD matrix low-rank approximation [MM16, MW17].

We show that for the alternative kernel PCA guarantee of (2), a potentially faster runtime of $O(\mathrm{nnz}(A)) + \mathrm{poly}(n, k, 1/\sigma_{k+1}, 1/\epsilon)$ can be achieved for general shift and rotation-invariant kernels. Practically, improving the second term in our runtime, especially the poor dependence on $n$, is an important open question. Generally, computing the kernel matrix $K$ explicitly requires $O(n^2 d)$ time, and so our algorithm only gives runtime gains when $d$ is large compared to $n$ – at least $\Omega(n^{\omega-.5})$, even ignoring $k$, $\sigma_{k+1}$, and $\epsilon$ dependencies. Theoretically, removing the dependence on $\sigma_{k+1}$ would be of interest, as it would give input sparsity runtime without any assumptions on the matrix $A$ (i.e., that $\sigma_{k+1}$ is not too small). Resolving this question has strong connections to finding efficient kernel *subspace embeddings*, which approximate the full spectrum of $K$.

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
