[Supplementary Material · supplemental.pdf]

# A Fast low-rank approximation of $AA^T$

We give an algorithm which matches the lower bound of Theorem 3.

**Theorem 10.** *There is an algorithm, which given $A \in \mathbb{R}^{n \times d}$ computes $N \in \mathbb{R}^{n \times k}$ in $O(\mathrm{nnz}(A)k) + n \cdot \mathrm{poly}(k/\epsilon)$ time such that probability $99/100$:*

$$\|AA^T - NN^T\|_F^2 \leq (1+\epsilon)\|AA^T - (AA^T)_k\|_F^2.$$

*Proof.* It is known (see Lemma 11 of [CW17]) that there exists a distribution over random matrices $R, S \in \mathbb{R}^{n \times O(k/\epsilon)}$ which can be applied to $A$ in $O(\mathrm{nnz}(A)) + n \cdot \mathrm{poly}(k/\epsilon)$ time such that with probability $199/200$, setting

$$Y^* = \underset{Y \in O(k/\epsilon) \times O(k/\epsilon) \text{ with rank } k}{\arg\min} \|AA^T RY S^T AA^T - AA^T\|_F^2$$

we have:

$$\|AA^T RY^* S^T AA^T - AA^T\|_F^2 \leq (1+\epsilon)\|AA^T - (AA^T)_k\|_F^2.$$

We can solve for an approximately optimal $\tilde{Y}$ by further sketching our problem on the left and right (similar to the technique used in Lemma 15 of [CW17]). Specifically, if we let $T_L, T_R \in \mathbb{R}^{n \times \mathrm{poly}(k/\epsilon)}$ be drawn from the Count Sketch distribution, we can solve:

$$\tilde{Y} = \underset{Y \in O(k/\epsilon) \times O(k/\epsilon) \text{ with rank } k}{\arg\min} \|T_L^T AA^T RY S^T AA^T T_R - T_L^T AA^T T_R\|_F^2$$

and are guaranteed that with probability $99/100$,

$$\|AA^T R\tilde{Y} S^T AA^T - AA^T\|_F^2 \leq (1+2\epsilon)\|AA^T - (AA^T)_k\|_F^2. \tag{11}$$

Computing $\tilde{Y}$ requires forming $T_L^T A$, $A^T R$, $S^T A$, and $A^T T_R$ and then multiplying the appropriate matrices together. This takes $O(\mathrm{nnz}(A)) + n \, \mathrm{poly}(k/\epsilon)$ time. Once $T_L^T AA^T R$, $S^T AA^T T_R$ and $T_L^T AA^T T_R$ have been formed we can solve for $\tilde{Y}$ in $\mathrm{poly}(k/\epsilon)$ time using the formula of [FT07].

Finally, since $\tilde{Y}$ is rank-$k$ we can factor $\tilde{Y} = VV^T$ for $V \in \mathbb{R}^{O(k/\epsilon) \times k}$ using the SVD. We can then compute $N_1 = AA^T RV \in \mathbb{R}^{n \times k}$ and $N_3 = AA^T SV \in \mathbb{R}^{n \times k}$ which satisfy $\|AA^T - N_1 N_2^T\|_F^2 \leq (1+2\epsilon)\|AA^T - (AA^T)_k\|_F^2$ with probability $99/100$ by (11).

$N_1$ and $N_2$ both require $O(\mathrm{nnz}(A)k) + n \cdot \mathrm{poly}(k/\epsilon)$ time to compute. The theorem follows from adjusting constants on $\epsilon$ and noting that we can symmetrize $N_1 N_2^T$ to form $NN^T$ if desired in $n \cdot \mathrm{poly}(k/\epsilon)$ time. $\qquad\square$

# B Hardness of outputting a low-rank subspace

Theorem 3 shows a lower bound on outputting a relative-error low-rank approximation to $MM^T$. Here we show that this hardness extends to the possibly easier problem of just outputting a low-rank span that contains a relative-error low-rank approximation. This result extends analogously to the other kernel lower bounds discussed in Section 2.

**Theorem 11** (Hardness of low-rank span for $MM^T$). *Assume there is an algorithm $\mathcal{A}$ which given any $M \in \mathbb{R}^{n \times d}$ returns orthonormal $Z \in \mathbb{R}^{n \times k}$ such that $\|MM^T - ZZ^T MM^T\|_F^2 \leq \Delta_1 \|MM^T - (MM^T)_k\|_F^2$ in $T(M, k)$ time for some approximation factor $\Delta_1$.*

*For any $A \in \mathbb{R}^{n \times d}$ and $C \in \mathbb{R}^{d \times k}$ each with integer entries in $[-\Delta_2, \Delta_2]$, let $B = [A^T, wC]^T$ where $w = 3\sqrt{\Delta_1}\Delta_2^2 nd$. It is possible to compute the product $AC$ in time $T(B, 2k) + \tilde{O}((n + d)k^{\omega-1})$.*

*Proof.* $ZZ^T MM^T$ is the projection of $M$ onto the column span of $Z$. This projection can be performed approximately using standard leverage score sampling techniques (see e.g., [CW13]). Let $S \in \mathbb{R}^{s \times n}$ be a sampling matrix sampling rows of $Z$ by its row norms (its leverage scores since

it is orthonormal) where $s = c(k \log k)$ or some sufficiently large constant $c$. Let $R$ be the $n \times k$ matrix which selects the last $k$ columns of $MM^T$.

Letting $X^* = \arg\min_{X \in k \times k} \|ZX^T - MM^T R\|_F^2$ and $X = \arg\min_{X \in k \times k} \|SZX^T - SMM^T R\|_F^2$ we have by a well known leverage score approximate regression result with high probability in $k$:

$$
\begin{aligned}
\|ZX^T - MM^T R\|_F^2 &= O(1) \cdot \|Z(X^*)^T - MM^T R\|_F^2 \\
&= O(1) \cdot \|ZZ^T MM^T R - MM^T R\|_F^2 \\
&= O(\Delta_1) \|MM^T - (MM^T)_k\|_F^2.
\end{aligned}
$$

Further, computing $X$ requires $\tilde{O}(dk^{\omega-1})$ time to compute the $O(k \log k) \times k$ submatrix $SMM^T R$ as well as $\tilde{O}(k^\omega) = \tilde{O}(nk^{\omega-1})$ to perform the regression. This gives the result via Theorem 3 since computing $Z$ with rank-$2k$ $ZX^T$ gives a low-rank approximation of $MM^T$ with error $O(\Delta_1) \|MM^T - (MM^T)_{2k}\|_F^2$ *measured on the last $k$ columns of $M$*. Small error on these columns is all that is needed to recover $AC$ accurately (see proof of Theorem 3). $\qquad\square$

## C   Additional lower bound proofs

We now prove our hardness result for kernels depending on the squared distance $\|a_i - a_j\|_2^2$.

**Theorem 5.** *Consider any kernel function $\psi : \mathbb{R}^d \times \mathbb{R}^d \to \mathbb{R}^+$ with $\psi(a_i, a_j) = f(\|a_i - a_j\|^2)$ for some function $f$ which can be expanded as $f(x) = \sum_{q=0}^\infty c_q x^q$ with $c_1 \neq 0$ and $|c_q/c_1| \leq G^{q-1}$ and for all $q \geq 2$ and some $G \geq 1$.*

*Assume there is an algorithm $\mathcal{A}$ which given input $M \in \mathbb{R}^{n \times d}$ with kernel matrix $K = \{\psi(m_i, m_j)\}$, returns $N \in \mathbb{R}^{n \times k}$ satisfying $\|K - NN^T\|_F^2 \leq \Delta_1 \|K - K_k\|$ in $T(M, k)$ time.*

*For any $A \in \mathbb{R}^{n \times d}$, $C \in \mathbb{R}^{d \times k}$, with integer entries in $[-\Delta_2, \Delta_2]$, let $B = [w_1 A^T, w_2 C]^T$ where $w_1 = \frac{w_2}{36\sqrt{\Delta_1}\Delta_2^2 nd}$ and $w_2 = \frac{1}{(16Gd^2\Delta_2^4)(36\sqrt{\Delta_1}\Delta_2^2 nd)}$. It is possible to compute $AC$ in time $T(B, 2k+3) + O(nk^{\omega-1})$.*

*Proof.* Define the distance matrix $D \in \mathbb{R}^{n+k \times n+k}$ with $D_{i,j} = \|b_i - b_j\|^2$. Using the fact that $\|b_i - b_j\|^2 = \|b_i\|^2 + \|b_i\|^2 - 2b_i^T b_j$ we have $D = E + E^T - 2BB^T$ where $E$ is a rank-1 matrix with all rows equal to $[\|b_1\|_2^2, ..., \|b_{n+k}\|_2^2]$. We can write the kernel matrix for $B$ and $k$ as:

$$
K = c_0 \begin{bmatrix} 1 & 1 \\ 1 & 1 \end{bmatrix} + c_1(E + E^T) - 2c_1 \begin{bmatrix} w_1^2 AA^T & w_1 w_2 AC \\ w_1 w_2 C^T A^T & w_2^2 C^T C \end{bmatrix} + c_2 D^{(2)} + c_3 D^{(3)} + ... \quad (12)
$$

where $D_{i,j}^{(q)} = \|b_i - b_j\|^{2q}$. Let $\bar{K}$ be $K - c_0 \cdot 1 - c_1(E + E^T)$, with its top $n \times n$ block set to $0$. $\bar{K}$ has rank at most $2k$ and if we set $Q \in \mathbb{R}^{n \times 2k+3}$ to be a matrix with columns spanning the columns of $\bar{K}$, the all ones vector, $E$ and $E^T$, then letting $N$ be the result of running $\mathcal{A}$ on $B$ with rank $2k + 3$:

$$
\|K - NN^T\|_F^2 \leq \Delta_1 \|K - QQ^T K\|_F^2 \leq \Delta_1 \left\| \begin{bmatrix} -2c_1 w_1^2 AA^T + c_2 \hat{D}^{(2)} + ... & 0 \\ 0 & 0 \end{bmatrix} \right\|_F^2 \quad (13)
$$

where $\hat{D}^{(q)}$ denotes the top left $n \times n$ submatrix of $D^{(q)}$.

By our bounds on the entries of $A$, for $i, j \leq n$, $\|b_i - b_j\|^2 \leq 4d\Delta_2^2 w_1^2$ and by our setting of $w_1, w_2$, plugging into (13) we have for all $i, j$:

$$
|(K - NN^T)_{i,j}| \leq \|K - NN^T\|_F \tag{14}
$$

$$
\leq \sqrt{\Delta_1} n \left( 2c_1 d\Delta_2^2 w_1^2 + \sum_{q=2}^\infty c_q (4d\Delta_2^2 w_1^2)^q \right)
$$

$$
\leq \sqrt{\Delta_1} n c_1 d\Delta_2^2 w_1^2 \left( 2 + \sum_{q=2}^\infty (4Gd\Delta_2^2 w_1^2)^{q-1} \right) \qquad \text{(Since } |c_q/c_1| \leq G^{q-1})
$$

$$
\leq 3\sqrt{\Delta_1} n c_1 d\Delta_2^2 w_1^2 \leq \frac{w_1 w_2 c_1}{12} \tag{15}
$$

where the second to last bound follows from the fact that $w_1 < w_2$ and $w_2$ is set small enough so $(4Gd\Delta_2^2) \cdot w_2^2 \ll 1/2$ so the series converges to a sum $< 1$. Additionally, for $i \leq n$ and $j \leq k$ (i.e., considering the entries of $K$ corresponding to AC) we have:

$$K_{i,n+j} = c_0 + c_1(E + E^T)_{i,n+j} - 2c_1 w_1 w_2(AC)_{i,j} + \sum_{q=2}^{\infty} c_q D_{i,n+j}^{(q)}.$$

This last sum can be bounded by:

$$\left| \sum_{q=2}^{\infty} c_q D_{i,n+j}^{(q)} \right| \leq c_1 \sum_{q=2}^{\infty} G^{q-1}(4\Delta_2^2 dw_2^2)^q \qquad \text{(By assumption } |c_q/c_1| \leq G^{q-1})$$

$$\leq c_1 w_1 w_2 \sum_{q=2}^{\infty} G^{q-1} w_2^{2(q-1)} \frac{w_2}{w_1} \left(4\Delta_2^2 d\right)^q$$

$$\leq c_1 w_1 w_2 \sum_{q=2}^{\infty} G^{q-1} w_2^{2q-3} \left(4\Delta_2^2 d\right)^q \qquad \text{(Using } \frac{w_2}{w_1} \leq \frac{1}{w_2}.)$$

$$\leq \frac{1}{3} c_1 w_1 w_2. \qquad \text{(Using } w_2 \leq \frac{1/4}{16G\Delta_2^4 d^2} \text{ so the series converges.)}$$

If we set $v = NN_{i,n+j}^T - c_0 - c_1(E + E^T)_{i,n+j}$ we thus have combining with (14) for $i \leq n, j \leq k$

$$|v + 2c_1 w_1 w_2(AC)_{i,j}| \leq \frac{5c_1 w_1 w_2}{12}$$

and so we can compute $(AC)_{i,j}$ exactly by rounding $v$ to the nearest integer multiple of $c_1 w_1 w_2$. This gives the theorem since we can compute the required entries of $NN^T$ and $E$ in $O(nk^{\omega-1})$ time. □