[Reviews · NeurIPS 2017]

Reviewer 1



The authors study the problem of k-rank approximation of kernel matrix K has a complexity of at least \Omega(nnz(A)k), where nnz(A) is the number of non-zero elements in input data. The authors also show that for some classes of kernels kernelized dataset can be low-rank-approximated in O(nnz(A)) time. The focuses purely to the theory of kernel approximations and it contains no real dataset or éven a short discussion as to how this would advance the current state of the art or conclusion with prosects of practical implications of the results. The paper is written quite concisely. I am not 100% confident that I am able to review the significance of this theoretical contribution, but I would have liked some connection to real applications as well. The citation style is not pursuant to the NIPS guidelines (the manuscript uses citations of type [SS00] while the NIPS style would be [1] etc.).

Reviewer 2



The paper presents negative and positive results to the problem of computing low-rank approximations to the kernel matrix of a given dataset. More specifically, for a fixed kernel function and given some input data and a desired rank k, the authors consider the two related problems of (1) computing a rank k matrix which is relatively close to the optimal rank k approximation of the kernel matrix and of (2) computing an orthonormal basis of a k-dimensional space such that when projecting the data from the feature space (defined by the kernel) down to this k-dimensional space it is relatively close to the optimal such projection. In particular, the paper focuses on the question whether it is possible to solve these problems in a time that is independent of the dimensionality of the input data matrix A (say n times d) and instead depends on the number of non-zero entries in A. Regarding problem (1) the paper provides a negative answer for the linear kernel, all polynomial kernels, and the Gaussian kernel by showing that the problem is as hard as computing (exactly) the product of the input matrix with an arbitrary matrix C (of dim d times k). This implies that a solution cannot be achieved without a significant improvement of the state of the art in matrix multiplication. Regarding problem (2) the paper provides a positive result for shift-invariant kernels using an algorithm based on random Fourier features. While the practical relevance of the given algorithm is limited due to a poor scaling with the number of data points (n to the 4.5), its existence is theoretically significant because it achieves the desired independence of n times d and, thus, gives hope for further improvement down to a practically relevant dependencies on n. The given results constitute valuable and technically challenging contributions to a topic of high relevance to the NIPS community. Additionally, the paper is excellently written including a good exposition of the results and high-level outline of the proofs, which makes it easier to follow the detailed steps.

Reviewer 3



This paper proposes a method to make a low-rank (rank k) approimation of a (n*n) kernel matrix which is faster when the dimensionality of the dataset d is very high (d >> n) and the input data A is very sparse. Understanding when and for what goal the contribution is useful is non-trivial. The paper says high-dimensional data often appears (which is true) and cites genetics data, but often genetics data is not extremely sparse (e.g. lists for every common mutation point the variant observed, where often more than one variant is non-rare). Even so, if the feature space is high-dimensional, one often will choose algorithms which are only linear in the dimension d (and have a higher complexity in the smaller n). Most operations are possible in O(n^3) (e.g. matrix multiplication, matrix inversion, ...). So is low-rank approximation really needed if n is "not too large" ? If one can afford the proposed algorithm with complexity O(n^{4.5}...), I would expect low-rank approximation is not needed in many cases. Even so, the O(n^{4,5}) complexity seems to imply that the number of data points in the input data should be rather small, which is a clear limitation of the proposed algorithm. The paper doesn't show experiments. This is not needed in a lot of analytic work, but here I'm left with the practical question of what would be a concrete task where the proposed approach would be better than "no low rank approximation", and would also be better than "classic low rank approximation strategies". The paper doesn't have a conclusions/further work section, which is useful to summarize and evaluate, e.g. list strong / weak points and how one could mitigate the weaknesses. DETAILS: Line 89: be extending -> be extended Line 91: remove duplicate "achieving" Line 104: accept -> except